# Utilizing a Building Information Modelling Environment to Communicate the Legal Ownership of Internet of Things-Generated Data in Multi-Owned Buildings

**Behnam Atazadeh ***[ID]**, Hamed Olfat**[ID]**, Behzad Rismanchi**[ID]**, Davood Shojaei**[ID] **and Abbas Rajabifard**

Centre for Spatial Data Infrastructures and Land Administration, Department of Infrastructure Engineering, The University of Melbourne, Parkville, VIC 3010, Australia; olfath@unimelb.edu.au (H.O.); behzad.rismanchi@unimelb.edu.au (B.R.); shojaeid@unimelb.edu.au (D.S.); abbas.r@unimelb.edu.au (A.R.)

\* Correspondence: behnam.atazadeh@unimelb.edu.au

**Abstract:** In multi-owned buildings, a community of residents live in their private properties while they use and share communal spaces and facilities. Proper management of multi-owned buildings is underpinned by rules related to health, safety, and security of the residents and visitors. Utilizing Internet of Things (IoT) devices to collect information about the livable space has become a significant trend since the introduction of first smart home appliances back in 2000. The question about who owns the IoT generated data and under what terms it can be shared with others is still unclear. IoT devices, such as security camera and occupancy sensors, can provide safety for their owners, while these devices may capture private data from the neighborhood. In fact, the residents are sometimes not aware of regulations that can prevent them from installing and collecting data from shared spaces that could breach other individuals' privacy. On the other hand, Building Information Modelling (BIM) provides a rich 3D digital data environment to manage the physical, functional, and ownership aspects of buildings over their entire lifecycle. This study aims to propose a methodology to utilize BIM for defining the legal ownership of the IoT generated data. A case study has been used to discuss key challenges related to the ownership of IoT data in a multi-owned building. This study confirmed that BIM environment can facilitate the understanding of legal ownership of IoT datasets and supports the interpretation of who has the entitlement to use the IoT datasets in multi-owned buildings.

**Keywords:** IoT data; BIM; multi-owned buildings; ownership spaces

## 1. Introduction

Building Information Modelling (BIM) is a design process that is used in current Architecture, Engineering and Construction (AEC) practices to describe, model, and publish the documentation required for constructing buildings, structures and urban design in general [1]. It is becoming a dominant paradigm to provide a spatially, temporally and semantically accurate multi-dimensional data environment for facilitating communication and collaboration in the AEC industry. On the other hand, Internet-of-Things (IoT) is typically defined as a system of interconnected sensing and computing devices or things with unique identifiers, enabling intelligent communication of data over a unified network and eliminating the need for human-to-human or human-to-computer interaction. IoT focuses on enabling communication between all devices, things that are existent in real life or that are virtual [2]. IoT is not just concerned with devices such as sensors and actuators, but more importantly, the fundamental aspect of IoT is the interaction of devices via an internet-based environment.

The integration of BIM with real-time data from IoT devices presents a powerful paradigm for applications to improve construction and operational efficiencies [3]. Currently, a wide range of IoT devices and sensors are being deployed for various applications such as monitoring air quality, noise, temperature, security and energy consumption. This deployment is rising dramatically, and it is expected that approximately 50 billion IoT devices and sensors will be installed across the globe by 2020 [4,5]. The vast majority of IoT devices were deployed in the built environment. In order to unlock the value of IoT datasets for the built environment, these datasets should be linked with BIM models to visually communicate and represent them in a real-world context. In other words, to enable BIM as a real-time information repository of the built environment, datasets coming from IoT nodes and sensors/sensor networks need to be incorporated into BIM. The sensors that are monitoring every building element (when integrated with data objects of the BIM) will provide meaningful information about the states of the building elements and also regarding the states of the spaces (rooms, corridors, etc.) in the building. Several conditions in the built environment such as temperature, humidity, gas concentration, sound level, occupant count, motion detection and many more attributes can be measured/monitored in real-time over IoT platforms [2].

One major challenge of IoT generated dataset is associated with the legal ownership of the measured attributes. It is unclear who owns the data collected by the sensors and to what extent it is legitimate to capture data in a built environment. A common type of buildings in urban areas is multi-owned buildings. In these buildings, owners of individual units live together with shared rights and responsibilities for communal parts (known as common property) of the building [6]. In multi-owned buildings, deployment of IoT devices can help owners to monitor their private and common areas. For example, owners are typically members of owners corporations who are responsible for managing common facilities. According to their liabilities, they need to pay a certain amount of expenses to the owners corporation for curating and maintaining common facilities, such as monitoring energy consumption in corridors and lobby areas.

The management of private and common properties can be more efficient if IoT devices are installed in various parts of multi-owned buildings. However, the spatial extent of IoT coverage spaces is sometimes intervene with the legal spaces that define ownership boundaries of properties. In other words, an IoT sensor or device may capture the data related to private or communal parts of a multi-owned building without the consent of residents holding legal entitlements over those parts. This may lead to privacy breaches and creating legal issues if IoT devices that are not appropriately deployed in a multi-owned building. A simple example could be a shared parking lot, if the owner of the parking lot "A" installs a security camera to observe the activities around their car and the installed camera has oversights to the parking lot "B", the captured data could reveal the privacy of the owner of the parking lot "B". Figure 1 illustrates the potential data privacy breach.

The aim of this study is to propose a BIM-driven approach to simultaneously represent coverage spaces of IoT devices as well as ownership boundaries of private and common properties in a 3D data environment. A 3D BIM model for a multi-owned building was implemented and various IoT coverage spaces and ownership spaces were visually represented to showcase the viability of the proposed approach.

In the next section, the literature related to BIM and IoT integration to support smart management of the built environment is reviewed. Section 3 is dedicated to the proposed conceptual BIM model to map the spatial extent of IoT coverage spaces and property ownership spaces. This is followed by the presentation of a multi-owned building as a suitable case to demonstrate the proposed BIM-based approach in Section 4. Section 5 includes a discussion on using BIM environment to communicate issues related to legal ownership of IoT generated datasets and highlights legal challenges of prime importance when deploying IoT sensors in multi-owned properties. In the final section, conclusions, as well as future research directions, are presented.

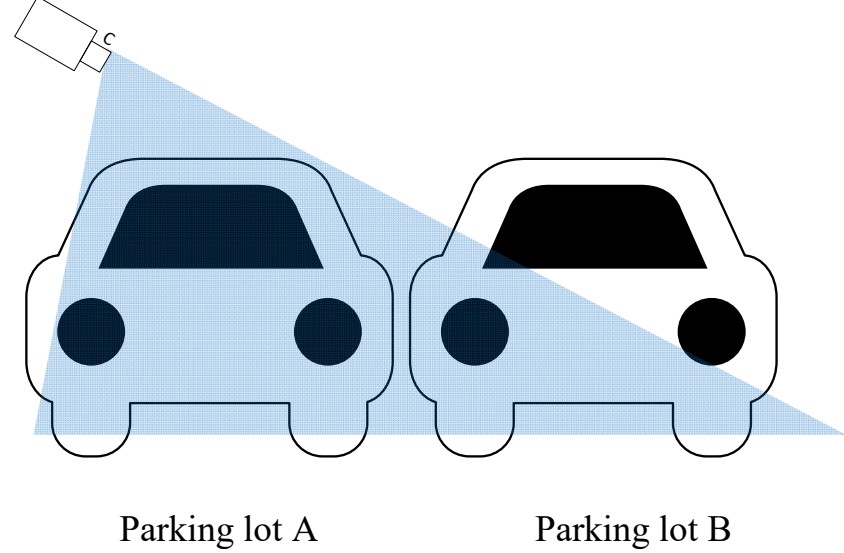

Parking lot A          Parking lot B

**Figure 1.** The potential privacy breach by an IoT based security camera installed for the parking lot "A" with an oversight on the parking lot "B".

## 2. Review of Relevant Literature

Digital Twins emergence is an endeavor to create intelligent adaptive machines by generating a parallel virtual version of the system along with the connectivity and analytical capabilities enabled by IoT [7]. The Digital Twin technology is moving towards creating a digital representation of a real-world object (e.g., a car, a machine tool, a factory, a person) and the spatio-temporal relations between the objects represented by a subgraph of nodes and edges [8]. Figure 2 illustrates this concept. For example, a car "T37BTT" is represented by multiple nodes and edges in a subgraph, of which its nodes represent the car's CAD design, the service records, its current state (where it is, its speed, etc.), and its manufacturing information (where it was produced, by which machines, etc.).

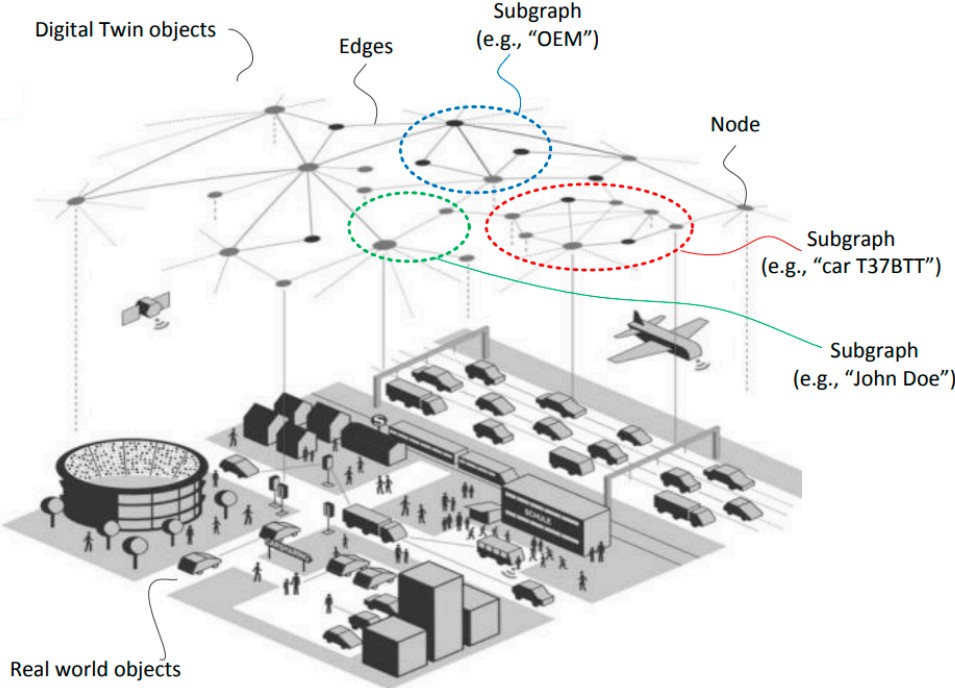

**Figure 2.** IoT Lifecycle via Digital Twins, adopted from [8].

Building objects in Digital Twin can be represented using 3D digital models. Figure 3 illustrates a BIM IFC file existing in the Victorian Government Digital Twin platform. It enables the Digital Twin users to access different physical elements of the building (e.g., ceiling, wall, roof, window) as well as its cadastral (title boundaries) information [9–12]. Similar to the abovementioned car "T37BTT" example, the nodes of a building can consist of CAD floor plans, utility map, occupancy/ownership/spaces/ temperature information, and camera footage.

Several researchers have already worked on integrating BIM and IoT; however, this research domain is still in nascent stages [3]. Isikdag [2] proposed an architecture, based on GIS, for integration and visualization of information coming from BIM objects and IoT nodes. Rowland [1] proposed an informational city model including BIM geometry, real-time data via IoT, multiplayer online gaming based platform and an Augmented Reality interface. Teizer et al. [13] focused on an IoT approach that integrates environmental and localization data in a cloud-based BIM platform, aiming at making performance, environmental and localization data of workers available in an indoor work environment. Dave et al. [14] proposed a framework for integrating BIM and IoT using open standards to provide information about energy usage, occupancy and user comfort. Tanga et al. [3] provided an in-depth review of BIM and IoT devices integration in the AEC industry from domain application and integration methodologies perspectives.

Smart buildings are predicated on the appropriate integration of IoT datasets and 3D BIM models. IoT devices could collect data during the construction (structure monitoring) and operation (energy meters). The purpose of these devices is to monitor the health and safety of the building which will benefit all occupants. Integration of BIM and IoT has been adopted and used for various applications. These include construction operation and monitoring [15,16], health and safety management [17,18], construction logistics and management [19,20], facility management [21,22], energy management [23], and disaster and emergency response services [24,25]. Here, we will review the recent investigations and initiatives that studied the integration of IoT datasets and 3D BIM models for supporting smart building management in different application domains.

In the construction operation and monitoring domain, the integration of BIM and IoT datasets provides new opportunities to integrate real-time datasets such as environmental data and localization data to support management and operations of construction activities. Since incorporating data sourced from IoT sensors into BIM models can facilitate real-time exchange and communication of data, monitoring construction activities would benefit from this in several aspects including on-site monitoring of the construction environment [26], monitoring of resources and labor behaviors [27], real-time communication and collaboration [28], and monitoring progress and performance of construction activities [29,30].

In terms of health and safety management, IoT datasets are considered valuable for smart monitoring of activities related to health and safety aspects. On the other hand, BIM models include a rich repository of information related to building components, which provide the right context for the datasets sourced from IoT devices. Two major applications of integrating BIM and IoT datasets for managing health and safety aspects include development of health and safety training systems to track the location of trainers, trainees, materials and equipment [31,32], and on-site monitoring of health and safety by providing real-time data query, identifying risks as well as visualizing and notifying them inside the BIM models [18].

In construction logistics and management domain, the combination of BIM and IoT datasets provide significant improvements in automating prefabrication and lean construction. BIM and IoT devices such as RFID tags are useful for tracking, visualization and automatic assembly in prefabricated manufacturing. In terms of lean construction, BIM coupled with IoT can facilitate the assessment of work progress, constraints and productivity by providing a reliable basis for information flow. Various investigations showed that integration of IoT datasets and BIM models would enable real-time information sharing, fostering communication between humans and machines throughout the entire supply chain and the building lifecycle [13,33–35]. Despite the benefits, there are some limitations in the

studies that looked at the integration of BIM and IoT for construction logistics and management. These limitations include; 1) Information overload due to a large amount of data [20] 2) Frameworks and solutions are conceptual or prototypical ones that only work in the laboratory environment [20,33,35], and 3) Proposed prototypical solutions are not realizable in real-world projects due to the conservative attitudes in the construction industry [35,36].

In facility management, BIM and IoT integration can help building and facility managers with automatic approaches to maintain and operate buildings over their lifecycle. This integration can lead to useful 3D digital data environments that can facilitate current practices for operation and maintenance of buildings including access to real-time data, monitoring maintainability of building assets, creation and update of digital assets and managing facility spaces [37]. Applications of BIM and IoT integration in facility management include:

(1)　Identification of building components and track them inside the BIM environment using RFID tags [38,39]
(2)　Defining linkages between physical assets and their digital counterparts through connecting building management systems (BMS) and BIM [21,22]
(3)　Retrieval of real-time data and visualizing facility management issues using portable devices equipped with BIM tools and augmented reality technology [40,41].

In terms of energy management, current research on the integration of IoT devices with BIM environment has developed various methods for visual communication and monitoring of energy usage in various building and city levels, analyzing and benchmarking energy performance. These have been accomplished by proposing new BIM-driven solutions for energy management and the integration of IoT based wireless sensor networks. These solutions are categorized into five main categories:

(1)　Energy management solutions in a web-based environment that provides the capability to visualize BIM models, query data sourced from energy sensors and receive actuation suggestions on an approximately real-time basis [42–44]
(2)　Geospatial information systems (GIS) driven solutions for managing energy, which can represent 3D digital models and track energy usage in a geospatially referenced context [23]
(3)　System based solutions for energy management to provide a real-time tracking of energy usage/production and building conditions in a BIM environment [14,45,46]
(4)　Solutions for analyzing energy performance based on real-time energy usage/generation using simulation software such as Building Controls Virtual Test Bed and Cyber-Physical Building Energy Management System [47,48]
(5)　Integration of wireless sensor networks and BIM for tracking energy flow, receiving feedback as well as benchmarking and controlling energy in real-time [49–51].

BIM and different kinds of IoT devices can potentially help with the development of efficient and reliable platforms for disaster management and emergency response services on building, precinct and urban levels. In this context, BIM and IoT data integration have been used for various applications in emergency and disaster management. These include:

(1)　Indoor localization of residents, who trapped in a building, inside the BIM model [24]
(2)　Integration of building information from BIM models with the location data sensed from IoT sensors and residents' mobile devices to compute the shortest route for evacuation [25,52,53]
(3)　Development of mobile applications for evacuation using BIM-based APIs [54], and
(4)　Integration of BIM, GIS and IoT sensors to support emergency response services at a wide urban scale [46,55].

Despite significant developments on the integration of BIM and IoT datasets in the architecture, engineering and construction sectors, most of the current solutions have not considered the ownership

of IoT generated data inside the multi-owned buildings, in which various owners and stakeholders own private spaces and share and use communal spaces. Therefore, in this study, a BIM-based approach for simultaneous representation of ownership spaces and coverage spaces of IoT data is proposed by the authors.

## 3. The Proposed Approach

The proposed approach in this paper relies on the current data structure of the open IFC standard. Currently, the IFC standard provides entities for modelling physical building components and various functional spaces. In this study, the main architectural building elements that are important for the legal ownership of private and communal spaces in multi-owned buildings are considered. These physical elements include walls, doors, windows, ceilings, floors, and columns. In terms of functional spaces, two types of spaces are considered; 1) ownership spaces and 2) IoT coverage spaces. These functional spaces are modelled by the "IfcSpace" and "IfcExternalSpatialElement" entities in the IFC standard. It is also possible to group these spaces into zones (IfcZone). The approach consists of three main steps:

(1)　　Identifying relevant IFC entities for modelling the physical structure of buildings
(2)　　Modelling ownership spaces and boundaries in IFC
(3)　　Modelling IoT coverage spaces in IFC. Each step is explained in detail in the following subsections.

### 3.1. Identifying Relevant Entities for Physical Building Elements

All building objects in the IFC standard are defined as subtypes of 'IfcBuilding Element' entity. This entity and its subtypes are represented in Figure 4. Physical elements considered in this study are those that form the most primary components of a multi-owned building. These elements comprise walls (IfcWall), windows (IfcWindow), ceilings (IfcCeiling), floors (IfcFloor), doors (IfcDoor), and columns (IfcColumn). The geometric information about physical building elements in IFC is defined via their supertype 'IfcProduct'. 'IfcProduct' refers to two entities for defining geometry in BIM models, namely 'IfcObjectPlacement' and 'IfcProductRepresentation'. The 'IfcObjectPlacement' entity defines the coordinate system for all spatial objects. It provides the ability to define the coordinates either by referencing the world coordinate system or relative to another object in the BIM model.

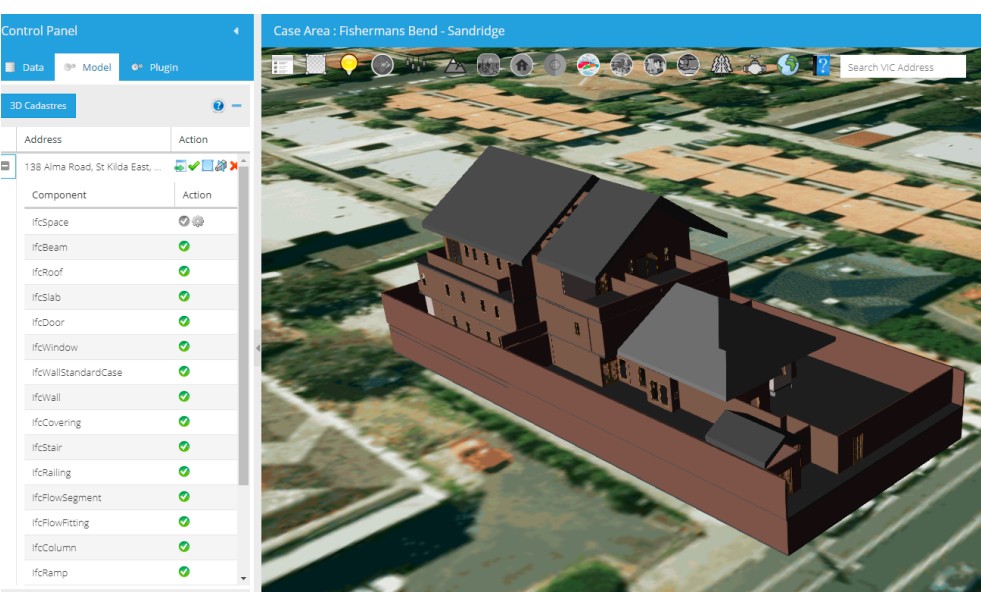

**Figure 3.** A building in IFC format within the Victorian Government Digital Twin platform.

*3.2. Modelling Ownership Spaces and Boundaries*

The relevant IFC entities for modelling ownership spaces and boundaries are represented along with physical building elements in Figure 4. For modelling ownership spaces and boundaries, the key entity is 'IfcRelSpaceBoundary' which defines the connection between physical elements (IfcBuildingElement) and ownership spaces. There are two main entities for representing functional spaces in the IFC standard, namely 'IfcSpace' for indoor spaces and 'IfcExternalSpatialElement' for outdoor spaces. In this study, 'IfcSpace' is mainly used for modelling ownership spaces since the focus of this study is limited to the spaces inside multi-owned buildings. Ownership spaces are defined by referencing two types of boundaries: physical and virtual. Each type is described in details in the following subsections.

3.2.1. Modelling Ownership Spaces Defined by Physical Elements

For modelling ownership spaces referencing physical elements, 'IfcBuildingElement' (or its subtypes) should be referenced using the 'RelateBuildingElement' attribute defined in 'IfcRelSpaceBoundary' [56,57]. Furthermore, there is another attribute called 'IfcPhysicalOrVirtualEnum' that must have the value of PHYSICAL since ownership spaces are defined by referring to physical objects. The location of boundaries could be defined in various parts of a physical building element, including the internal face, median surface and external face. Therefore, there is another attribute called 'InternalOrExternalBoundary' which helps us differentiate internal boundaries from external ones. Nevertheless, there is no 'Median' value assigned to this attribute. Therefore, we proposed adding this value to the "InternalOrExternalBoundary" to determine median boundaries for building elements.

3.2.2. Modelling Ownership Spaces Defined by Virtual Elements

Virtual elements are those imaginary surfaces, such as those in balcony and parking areas, that are used for defining ownership spaces [58]. Therefore, virtual ownership boundaries are not physically manifested. There is an entity called 'IfcVirtualElement' (represented in grey color in Figure 4) that is used for modelling virtual elements. Inside the "IfcRelSpaceBoundary" entity, the 'RelatedBuildingElement' attribute should refer to the "IfcVirtualElement" entity. In addition, the value of 'IfcPhysicalOrVirtualEnum' attribute must be defined as VIRTUAL because these boundary types do not exist in physical reality. There is no need to define a value for 'InternalOrExternalBoundary' attribute.

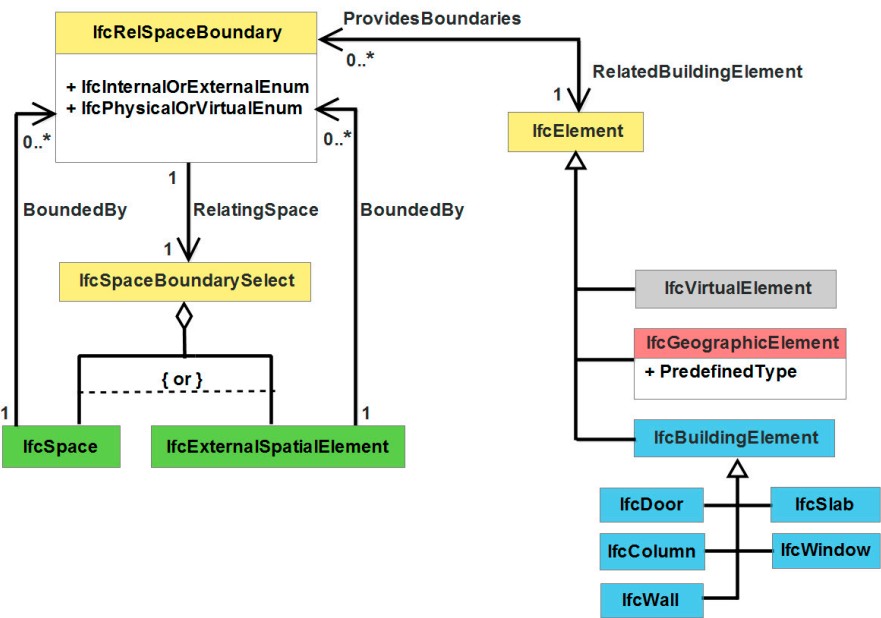

**Figure 4.** Entities for modelling building elements and functional spaces in the IFC standard.

### 3.2.3. Geometric Modelling of Ownership Boundaries in IFC

In addition to defining the boundary types, their geometry is also essential in BIM models. The geometry of ownership boundaries is defined by 'IfcConnectionGeometry' entity in the IFC (see Figure 5). This entity has a relationship with 'IfcRelSpaceboundary' using the 'ConnectionGeometry' attribute. As we consider that ownership spaces are in 3D and volumetric shape, the geometry of boundaries should be defined using surfaces or faces. In this context, the subtype entity 'IfcConnectionSurfaceGeometry' is adopted for geometric modelling of ownership boundaries between two spaces. This entity has an attribute called 'SurfaceOnRelatingElement' that defines the type of spatial connection between two ownership spaces. If the connection is defined a purely geometric surface, then the attribute refers to the 'IfcSurface' entity. If the connection is a geometric surface associated with a topological face, then the attribute refers to 'IfcSurfaceOrFaceSurface' entity. It should be noted that spatial placement of boundaries is specified by referring to their relating ownership space. In addition, there is another optional 'SurfaceOnRelatedElement' attribute that includes the same geometric modelling of the ownership boundary with reference to the related ownership space.

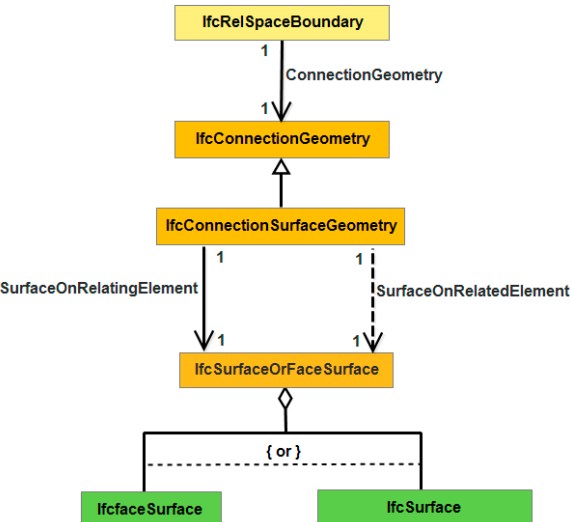

**Figure 5.** Geometric modelling of ownership boundaries in IFC.

### 3.3. Modelling IoT Coverage Spaces

There is a wide range of IoT devices and each IoT device type has its own specific coverage space. Some IoT sensor measures a point attributes such as temperature sensors, and some collect data from an area, such as movement sensors. Therefore, the geometric extent of the coverage space of a particular IoT device can be defined based on its specification. In this study, two typical examples of IoT devices is considered to showcase how IoT coverage spaces can be modelled inside the BIM environment.

The first example of an IoT device is the WiFi router. Typically, the coverage space of each WiFi router is considered as a sphere. It is very common in almost all multi-residential buildings to see the list of all neighboring WiFi routers. This is also applied to smart TVs and Bluetooth speakers. For instance, Figure 6 shows two WiFi coverage spaces, which are represented in dashed red and purple circles in a floorplan view, in two apartment units. Each WiFi coverage space is defined by an instance of 'IfcSpace' entity. In addition, the ownership space of each apartment unit is represented in green and blue colors, which are modelled by the instances of 'IfcSpace'. It also provides an example of how physically internal ownership boundaries are defined by a wall between two apartment units using IFC entities.

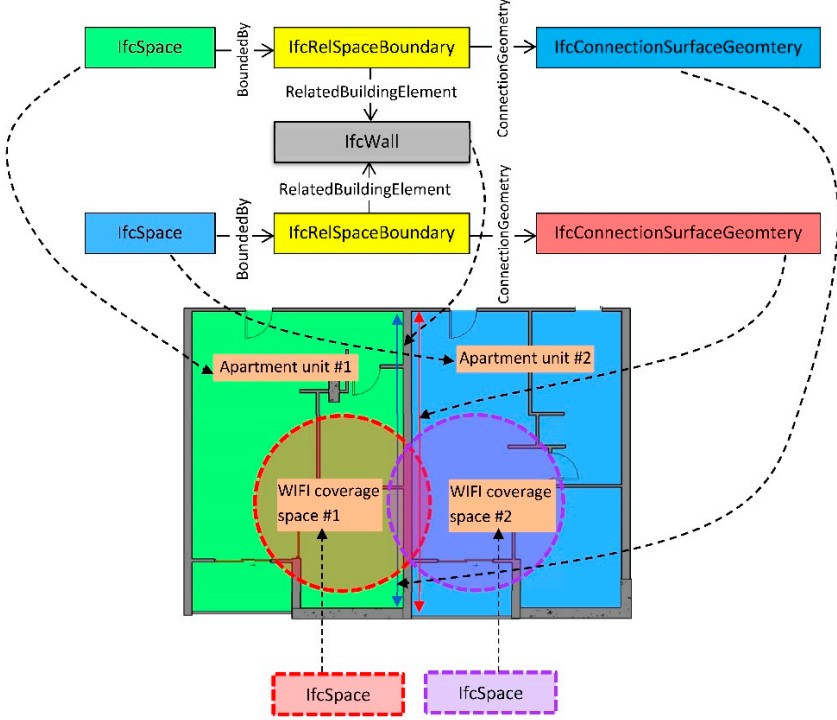

**Figure 6.** Modelling WIFI coverage and ownership spaces for two apartment units in IFC.

The second example refers to the CCTV cameras field of view. Similarly, instances of 'IfcSpace' entity can be used for modelling the geometric extent of CCTV cameras field of view. However, the coverage of CCTV cameras has a different shape compared to WiFi sensors. Figure 7 shows two CCTV cameras installed in two parking areas and their fields of view are mapped using 'IfcSpace.

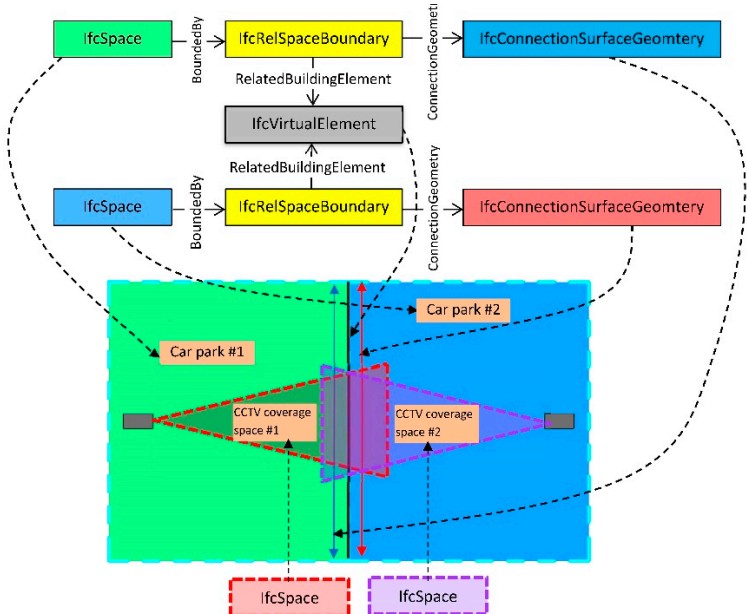

**Figure 7.** Mapping the field of view for two CCTV cameras and ownership spaces for two car parks in IFC.

These are represented in dashed purple and red triangles in a floorplan view. In addition, the ownership space of each car park is represented in green and blue colors, which are modelled by the

instances of 'IfcSpace'. It also provides an example of how ownership boundaries are defined by a virtual line between two car parks using IFC entities.

## 4. Case Study Implementation

In order to showcase the feasibility of the BIM and IFC standard for communicating ownership spaces and IoT coverage spaces in a common data environment, a BIM model for a multi-owned building has been constructed. The software package used in this study is Autodesk Revit, as a popular BIM authoring tool in the AEC industry. There are three steps considered in preparing the case study datasets:

(1) Creating physical elements of multi-owned buildings: 2D architectural CAD plans were used as the basis for creating the major building components. Revit has the ability to import these CAD plans and create architectural BIM components, such as walls, doors, windows, ceilings, floors and stairs, for existing buildings. Figure 8a show the entire physical model of the building.

(2) Defining ownership spaces and boundaries: In this step, ownership spaces and boundaries related to various privately-owned properties as well as common properties in multi-owned buildings were created. The basis of defining these ownership spaces and boundaries is the subdivision plans. Therefore, a subdivision plan of the case study building to create ownership spaces and boundaries is used. First, ownership boundaries were defined using two approaches. For physical boundaries, the 'Room Bounding' attribute defined for physical elements such as walls, ceilings and columns are used. By checking this attribute, the physical element is automatically considered as a boundary for space. The 'Room Separator' or 'Space Separator' tools can be used to define virtual boundaries in Revit. After defining boundaries, the 'Space' or 'Room' tool can be used to create each ownership space. A wide range of ownership spaces was created for private properties (such as apartment units, parking and storage areas) and common properties (such as corridors, elevator and stair areas, and driveways in carparks). Figure 8b shows all the ownership spaces together with physical elements of the building.

(3) Creating coverage spaces for the IoT devices: Coverage spaces for some IoT device examples were created to describe the geometric extent of each IoT device. In other words, these spaces show how much of the space an IoT device senses in the case study building. IoT coverage spaces have been defined using the 'Mass' tool in Revit. This tool provides the ability to define various spaces with different shapes depending on the type of the IoT device. For instance, spheres were created for representing the coverage of WiFi sensors while pyramids were used for describing the field of view for CCTV security cameras.

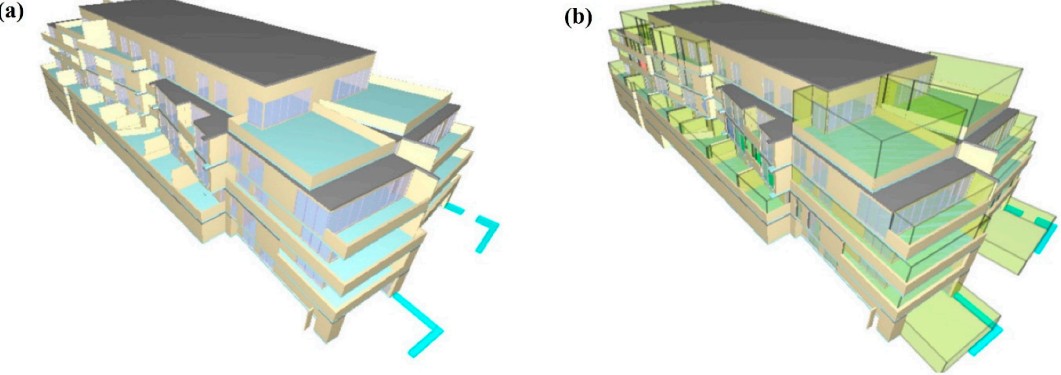

**Figure 8.** The case study multi-owned building: (**a**) Physical representation, (**b**) Ownership spaces integrated with the physical representation.

The prepared BIM model in the Revit format does not provide the ability to show ownership spaces in its 3D view. The BIM model was therefore converted into the standard format of IFC schema,

as described in Section 3.2, to better communicate ownership spaces and IoT coverage spaces in the 3D physical context of the building.

## 5. Discussion

Integration of IoT and BIM datasets brings various opportunities for managing the entire lifecycle of complex buildings. However, using IoT datasets in multi-owned buildings could lead to some issues related to the legal ownership of these datasets. Many owners and residents typically use private and communal spaces in multi-owned buildings, and it is essential to clearly understand the rights, restrictions and responsibilities associated with the use of data sourced from IoT devices in these buildings. The aim of this study is, therefore, to explore the viability of BIM to represent and communicate the legal ownership spaces and IoT coverage spaces in a common data environment with a reference to the physical reality. The BIM model implemented in this study provides some clear benefits for understanding issues related to the use of IoT data in multi-owned buildings.

Figure 9 shows ownership spaces of two private apartment units (unit 104 and unit 105) and coverage spaces of WiFi devices installed inside each apartment unit. It can be seen that the coverage area of each WiFi device intervenes with the ownership space of its adjacent private apartment unit as well as the corridor area. This signifies the fact that the coverage space of WiFi devices would have an impact on the health of residents. For instance, the owner or resident of apartment unit 104 has the right to install d the WiFi device inside the ownership space of their apartment unit. However, he/she is responsible for the potential impact of WiFi signals on the residents who live in the adjacent private properties such as apartment unit 105. Similarly, the WiFi signals emitting from apartment unit 105 could potentially lead to health issues in nearby residents such as those ones living in apartment unit 104.

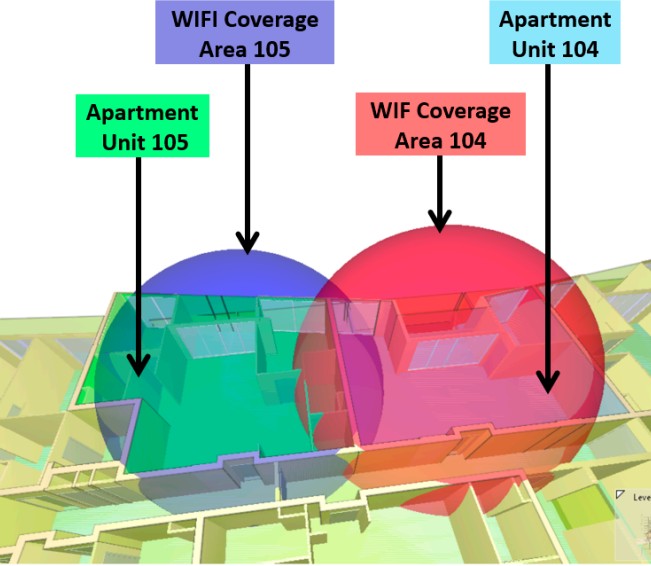

**Figure 9.** Two adjacent apartment units and their counterpart WiFi coverage spaces.

As another example shown in Figure 10, the owner of apartment unit 306 has installed a CCTV camera in front of their entry door to protect their private property. However, the security camera captures the corridor area which is a communal space. Communal spaces are typically jointly owned and used by all or a group of owners in a multi-owned building. Therefore, in this case, the owner breaches the privacy of other owners and residents who pass through the area covered by the CCTV camera. This could also bring some legal challenges in using CCTV cameras in various parts of a multi-owned building. Communal spaces or properties are typically managed by owners corporations and therefore the use of IoT devices in communal areas of a multi-owned building requires the consent

of all owners/users who are the member of owners corporation. Owners corporations have the full right and responsibility to access the IoT datasets captured in the communal spaces of a multi-owned building. The apartment owners are not entitled to use an IoT device, such as CCTV camera, in the communal area near to their entry doors unless it is approved by owners corporations.

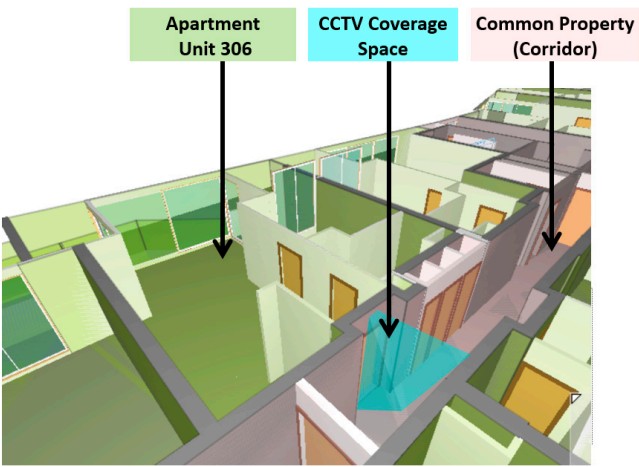

**Figure 10.** A CCTV coverage space represented inside the BIM model.

Figure 11 represents the ownership space of the apartment unit 205 and the space that is measured by an occupancy sensor. In this case, the occupancy sensor only measures the presence of a person inside some parts of the apartment unit 205. Since the occupancy space is within the ownership space of apartment unit 205, this situation does not bring legal issues to the use of occupancy sensor for this particular space. The owner of unit 205 is fully entitled to install and use the occupancy sensor inside their private properties.

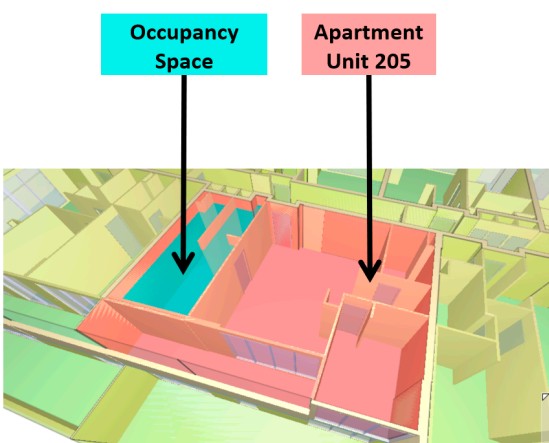

**Figure 11.** An example of the occupancy space that is covered by the ownership space.

In summary, the above examples give a demonstration of the legal issues related to the use of IoT devices in multi-owned buildings. The users may not be aware of these issues since there are no specific regulation or legislation to define how IoT datasets can be sourced and used by considering appropriate legal entitlements and liabilities. Integration of ownership spaces and coverage spaces of IoT devices in the BIM environment could help us to illustrate the use of IoT devices could breach the ownership spaces and these devices may capture data from private or communal spaces without the consent of owners and residents living in a multi-owned building.

Among various forms of privacy [59], territorial privacy refers to the invasion of ownership boundaries [60]. The approach proposed in this study is a type of territorial privacy preserving method

in multi-owned buildings. This approach could help with representation of various personal and communal territories, which are defined by ownership spaces, in multi-owned buildings and assist with protecting territorial privacy when utilizing IoT devices in these buildings.

## 6. Conclusions and Future Directions

This study has investigated the use of BIM environment to communicate rights, restrictions and responsibilities related to the use of IoT generated datasets in multi-owned buildings. The study showed that BIM could help with describing legal issues related to the dataset sourced from IoT devices if multiple owners and stakeholders are involved in complex buildings. In this study, various physical building and space elements inside the BIM standard (IFC) were identified to show how a BIM-driven approach could provide an integrated data structure to represent ownership spaces and IoT coverage spaces simultaneously and connect them to the physical reality of multi-owned buildings. Visualizing ownership spaces and IoT coverage spaces in the 3D digital data environment of BIM facilitates the understanding of legal ownership of IoT datasets and supports the interpretation of who has the entitlement to use the IoT datasets in multi-owned buildings. The scope of this study is limited to the multi-owned built environment. However, the suggested approach could be modified and applied in other sectors and application areas such as industrial, transportation, health and wellbeing, and building automation.

As a future research direction, it will be investigated how the integration of BIM and IoT datasets could help with managing common properties in multi-owned buildings. For instance, the owners should pay a certain amount of expenses to the owners corporation to maintain and manage communal services such as energy consumption. A BIM model integrated with IoT datasets could help monitor these costs in a more understandable and communicative way. Furthermore, BIM can provide a method for planning for the location of sensors in high-rises to provide the required security and efficiency. In addition, this integration can help residents to understand the location of sensors to consider their privacy. Last but not least, this integration of IoT and BIM provides a framework for having smart building concept which buildings react during the emergency situations such as incidents like fire, theft, and water leaking, by providing live and real-time datasets.

**Author Contributions:** Conceptualization, B.A. and B.R.; Methodology, B.A., H.O. and D.S.; Software, B.A.; Investigation, B.A., H.O., D.S., and B.R.; Data curation, B.A.; Writing—original draft preparation, B.A. and H.O.; Writing—review and editing, B.R. and D.S.; Visualization, B.A.; Supervision, A.R.; Project administration, A.R.; Funding acquisition, A.R.

**Funding:** This research was funded by Australian Research Council, grant number LP160100292.

**Acknowledgments:** This study was conducted as part of the Australian Research Council Linkage Project titled '3D Property Ownership Map Base for Smart Urban Land Administration'. The authors acknowledge the support of project partners: Land Use Victoria, Intergovernmental Committee on Surveying and Mapping (ICSM) and City of Melbourne. The authors emphasize that the views expressed in this article are the authors' alone.

**Conflicts of Interest:** The authors declare no conflict of interest. The funders had no role in the design of the study; in the collection, analyses, or interpretation of data; in the writing of the manuscript, or in the decision to publish the results.

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
