# Peer review of "Utilizing a Building Information Modelling Environment to Communicate the Legal Ownership of Internet of Things-Generated Data in Multi-Owned Buildings"

_electronics, doi:10.3390/electronics8111258_

Round 1

Reviewer 1 Report

This paper addresses an interesting topic overlapping the building information model and IoT-based sensing. Assumptions and discoveries were well explored and presented. Conclusions seem to be rather general considering the scope and significance, so  they could be enhanced if possible. Explanations on the BIM properties seem to a bit redundant, so it could be also improved. Otherwise, I have enjoyed reading the manuscript in general.

Author Response

Response to Reviewers' comments

The authors would like to thank the editor and the reviewers for their insightful comments and useful feedback, which have helped us to improve the quality of our manuscript. The detailed responses to your comments have been provided hereafter.

Reviewer 1:

This paper addresses an interesting topic overlapping the building information model and IoT-based sensing. Assumptions and discoveries were well explored and presented.

Conclusions seem to be rather general considering the scope and significance, so they could be enhanced if possible.

Response: The conclusion has been edited to capture more applications in the built environment.

Explanations on the BIM properties seem to a bit redundant, so it could be also improved.

Response:  Our explanations of the BIM could help readers with limited background in BIM to better understand the BIM concepts. We attempted to make these explanations as concise as possible throughout the manuscript.

Otherwise, I have enjoyed reading the manuscript in general.

Response: Thank you for the feedback

Reviewer 2 Report

In general,  this paper gives a good definition of the legal ownership of the IoT generated data which utilise BIM very well.

This paper has a good structure and clear flow, which are easy to follow, especially in the peer review part.

In the later stage, you can conduct to combine with the privacy fields, based on clear attribution ownership of the IoT data 

the below advice is just for the personnel view:

1. There is an excellent point for the multi-owned building, but also need to conduct the connection with the technical privacy-preserving methods, for instance, design a mechanism of privacy-preserving.

2. As for the IoT data, there is an effective way to create an entire project life cycle, including design and construction to operations and maintenance.

3. The sensor nodes are being deployed in various application areas such as the industrial, transportation, health and well being, building automation. It is also interesting to follow the other fields of the ownership.

Author Response

Response to Reviewers' comments

The authors would like to thank the editor and the reviewers for their insightful comments and useful feedback, which have helped us to improve the quality of our manuscript. The detailed responses to your comments have been provided hereafter.

Reviewer 2:

In general, this paper gives a good definition of the legal ownership of the IoT generated data which utilise BIM very well. This paper has a good structure and clear flow, which are easy to follow, especially in the peer review part.

In the later stage, you can conduct to combine with the privacy fields, based on clear attribution ownership of the IoT data the below advice is just for the personnel view:

Response: Thank you for the feedback

There is an excellent point for the multi-owned building, but also need to conduct the connection with the technical privacy-preserving methods, for instance, design a mechanism of privacy-preserving.

Response: In response to this comment, we identified four forms of privacy [1]:

Informational: It is associated with capturing and handling personal datasets Bodily: It concerns physical harms from invasive procedures Communications: It is related to any form of communication Territorial: This refers to the invasion of physical (ownership) boundaries [2]

Among these categories, our suggested approach is a type of a territorial privacy preserving method in multi-owned buildings. The BIM-driven approach could help with representation of various personal and communal territories, which are defined by ownership spaces, in multi-owned buildings and assist with protecting territorial privacy when utilizing IoT devices in these buildings. We added this explanation in the discussion section.

As for the IoT data, there is an effective way to create an entire project life cycle, including design and construction to operations and maintenance.

Response: The paragraph has been revised to clarify the context as follows:

Smart buildings are predicated on the appropriate integration of IoT datasets and 3D BIM models. IoT devices could collect data during the construction (structure monitoring) and operation (energy meters). The purpose of these devices is to monitor the health and safety of the building which will benefit all occupants. Integration of BIM and IoT has been adopted and used for various applications.

The sensor nodes are being deployed in various application areas such as the industrial, transportation, health and well being, building automation. It is also interesting to follow the other fields of the ownership.

Response:  In response to this comment, we added the below paragraph to the conclusion section:

The scope of this study is limited to the multi-owned built environment. However, the suggested approach could be modified and applied in other sectors and application areas such as industrial, transportation, health and wellbeing, and building automation.

References

Mendes, R.; Vilela, J. Privacy-Preserving Data Mining: Methods, Metrics and Applications. IEEE Access 2017, PP, 1. Könings, B.; Schaub, F.; Kargl, F.; Weber, M. Towards Territorial Privacy in Smart Environments. In Proceedings of the Intelligent Information Privacy Management (Privacy 2010), AAAI Spring Symposium; Stanford University, USA, 2010.
